# Gellan gum-based granular gels as suspension media for biofabrication

**Andrew McCormack[1]�, Laura M. Porcza[1]‡, Nicholas R. Leslie[1]‡, Ferry P. W. Melchels** [1,2]� *

1 Institute of Biological Chemistry, Biophysics and Bioengineering, School of Engineering and Physical Sciences, Heriot-Watt University, Edinburgh, United Kingdom, 2 Future Industries Institute, University of South Australia Mawson Lakes, Adelaide, Australia

he These authors contributed equally to this work.
‡ LMP and NRL also contributed equally to this work.
* ferry.melchels@unisa.edu.au

**Data Availability Statement:** All relevant data are within the paper and its Supporting information files.

**Funding:** AM EP/R513040/1 Engineering and Physical Sciences Research Council through their

## Abstract

Engineering 3D tissue-like constructs for applications such as regenerative medicine remains a major challenge in biomedical research. Recently, self-healing, viscoplastic fluids have been introduced as suspension media to allow lower viscosity, water-rich bioinks to be printed within them for the fabrication of more biomimetic structures. Here, we present gellan gum granular gels produced through the application of shear during gelation, as a candidate suspension medium. We demonstrate that these granular gels exhibit viscoplasticity over a wide range of temperatures, permitting their use for 3D bioprinting of filaments and droplets at low (4°C) as well as physiological temperatures. These granular gels exhibit very low yield stresses (down to 0.4 Pa) which facilitated printing at print speeds up to 60 mm·s⁻¹. Furthermore, we demonstrate the printing of cell-laden droplets maintained over 7 days to show the potential for multiple days of cell culture, as well as the fabrication of hydrogel features within a crosslinkable version of the suspension medium containing granular gellan gum and gelatine-methacryloyl. The combination of ease of preparation, high printing speed, wide temperature tolerance, and crosslinkability makes this gellan gum sheared through cooling-induced gelation an attractive candidate for suspended biofabrication.

## 1. Introduction

Granular hydrogels are emerging as a versatile platform for engineering tissue constructs in the biofabrication field. They offer an alternative microarchitecture for 3D cell culture compared to the conventional bulk polymer network hydrogels used ubiquitously in the field. Where the mechanical properties of bulk hydrogels arise solely from the cross-linking of polymer chains, the properties of granular hydrogels depend, to a large extent, on the degree of jamming of microparticles in their swollen state [1]. Around the jamming/unjamming transition of these microparticles, granular hydrogels exhibit unique physical properties such as a resistance to applied stress, shear-thinning and self-healing [1–3], making them suitable candidate materials for biofabrication applications. Specifically, these granular gels have recently been explored for 3D bioprinting, where they act as a supportive gel bath into which bioinks can be extruded [4–6].

Doctoral Training Programme https://www.ukri.
org/councils/epsrc/ The funders had no role in
study design, data collection and analysis, decision
to publish, or preparation of the manuscript.

**Competing interests:** The authors have declared
that no competing interests exist.

The use of suspension media with extrusion 3D printers is an emerging fabrication strategy that has been transformative to the 3D bioprinting field, primarily as it has removed the need to print layers sequentially, one on top of the other. With conventional layer-by-layer extrusion printing, optimising the rheological properties of bioinks to ensure a high degree of printability without loss of biological function is critical to the biofabrication process. Bioinks that provide a more biomimetic environment for cell encapsulation typically, have a high water-content and low viscosity [7, 8]. Such properties are detrimental to their printability by permitting rapid fluidisation of these inks post-printing, and thus collapse of printed structures [7]. Extruding low viscosity inks within the volume of a suspension media ensures support is given to the printed structure prior to secondary ink crosslinking mechanisms, while also facilitating printing both irregular and complex geometrical structures.

Notably, the main characteristics of suspension media that permit their ability to support extruded bioinks are related to their rheological properties [6]. These media are yield stress fluids, exhibiting a transition from an elastic solid to a viscoplastic fluid as their yield stress is overcome. As a result, the suspension media will fluidise through the action of the moving printing nozzle and can be easily displaced to accommodate the deposition of extruded ink [9]. Subsequently, these suspension media require a short recovery time to ensure complete immobilisation of the bioink following removal of the applied stress generated by the movement of the nozzle and the forces exerted by material deposition [9]. While the application of suspension media in biofabrication increases, the exploration of materials suitable for use as a suspension media is required to fully realise the value of this printing strategy for engineering of human tissues. The temperatures that a suspension media can be employed at are critical to their performance, often dictating the temperature range of the extruder nozzle [10], or maintaining the yield stress of the suspension media [4, 11]. We, therefore, looked at the development of suspension media that can be used over a wide range of temperatures. Furthermore, we hypothesised that using low concentrations of polymer in a granular gel would lead to a low yield stress, which may allow higher printing speeds to be obtained.

In this work we explored the preparation of granular gels using gellan gum and their application as suspension media in various biofabrication applications. Gellan gum is a biopolymer with a tetra-saccharide repeating unit consisting of glucose, glucuronic acid and rhamnose, and is already clinically used in the ophthalmic field in eye drop solutions [12]. The mechanism for the sol-gel transition of gellan gum involves conformational transition of the polymer chains in solution from a disordered random coil state to an ordered double helix state, followed by helix-helix aggregation [13]. In the presence of monovalent or divalent cations, the effect of electrostatic repulsions between helices is reduced, via the cations electrically shielding carboxylic groups on neighbouring polymer chains [14]. We identified gellan gum as a potential biopolymer for use as a suspension media, due to the polymer solutions exhibiting thermal gelation (~ 42–45˚C) and melting (~ 70–80˚C) temperatures much greater than physiological temperatures, thus preventing liquification of the polymer granules under cell culture of 37˚C.

Using gellan gum, we developed a rapid and reliable method for creating granular gel suspension media using polymer concentrations between 0.1–0.5% w/v, through applying shear as the gellan gum goes through its sol-gel transition upon cooling and in the presence of monovalent cations. This is a one-step method following the dissolution of the polymer, in contrast to other methodologies for producing granular gels which involve complete cooling-induced gelation followed by breaking up the macrogel into microgels, centrifugation and washing steps to produce a jammed granule network [4, 15]. Jammed granule networks exist due to a high granule-to-interstitial volume ratio, causing the gel to behave solid-like in the absence of an applied stress [1]. These gellan gum suspension media were used to demonstrate

the printing of filaments and cell-laden droplets. We further hypothesised that by combining these gellan gum granular suspension media with gelatine methacryloyl (gelMA), a medium would be obtained that retained both the suspension properties of the granular gel as well as the ability to photo-cure the resulting structures for long-term stability. This technique may allow the suspension medium to double as bulk matrix and allow for rapid fabrication of a larger tissue construct.

## 2. Materials and methods

### 2.1 Granular gel preparation

Known concentration of low acyl gellan gum (Alfa Aesar) was dissolved in phosphate buffered saline, PBS (Fisher Scientific). Following dissolution, the hot polymer was cooled through its sol-gel temperature (20 mins) under constant shearing of 600 RPM using an overhead stirrer (Caframo™ Petite Overhead Stirrer, Fisher Scientific). Prior to bioprinting granular gels were irradiated with a UV light source in a laminar biosafety cabinet for 30 min.

### 2.2 Photo-curable suspension media

Known concentration of gelMA (either 5 or 10% (w/v)) and 0.1% (w/v) lithium phenyl-2,4,6-trimethylbenzoylphosphinate (LAP, Sigma Aldrich) were added to a gellan gum granular gel (0.3% w/v) and dissolved at 37°C overnight. The resulting solution was then centrifuged for 2 min at 1000 RPM to remove air bubbles and stored at 4°C. Suspension media formulations were incubated at 37°C prior to use.

### 2.3 Polymer solutions used for biofabrication

3D printing was carried out using either poloxamer- or gelatine-methacryloyl (gelMA)–based solutions. Poloxamer solution was prepared by dissolving 30% (w/v) poloxamer 407 (Fisher Scientific) in deionised water at 4°C. In some instances, blue food dye was added to aid visualisation of printed filaments. GelMA-based solutions for droplet printing were prepared by dissolving 10% (w/v) gelMA in PBS with 0.05% (w/v) LAP photoinitiator. These solutions were heated to 50°C and then passed through a Whatman Puradisc 25 mm PES syringe filter, with a pore size of 0.2 μm prior to mixing in a cellular component. GelMA was prepared by reaction of type A gelatine (300 Bloom, Sigma Aldrich) derived from porcine skin tissue with methacrylic anhydride (Sigma Aldrich) for 1 h at 50°C. The experimental procedure was based on a previously described protocol developed by Van den Bulcke et al. [16]. Briefly, methacrylic anhydride was added dropwise to a 10% (w/v) solution of gelatine in phosphate-buffered saline (Fisher BioReagents), under constant stirring. Methacrylic anhydride was added at a ratio of either 0.02 g or 0.6 g per 1 g of gelatine. Before the addition of anhydride, the pH of the gelatine solution was adjusted to pH 8, with the addition of 5M NaOH solution. Following the reaction, centrifugation and dialysis (cellulose membrane, cut-off 12 kDa) against distilled water were performed to remove methacrylic acid and anhydride. The gelMA macromonomer solution was neutralised and then lyophilised and stored at -20°C until use. Validation of gelMA synthesis was carried out using proton-nuclear magnetic resonance spectroscopy ([1]H-NMR, Bruker AVIII 300 MHz).

GelMA solutions used for 3D printing of helix structures (FITC-gelMA) were prepared by dissolving 10% (w/v) fluorescein isothiocyanate (FITC) labelled gelMA in PBS with 0.1% (w/v) LAP photoinitiator.

## 2.4 Imaging and analysis

Granular gels were visualised by loading 200 μL onto a microscope slide mounted with an additional slide and imaged using an Eclipse TS100 inverted microscope (Nikon) equipped with a 10 X objective. Due to printed filaments requiring lower magnification for imaging, these structures were imaged using a digital camera (iPhone XS). All image processing was done using ImageJ. For droplet imaging, droplet surface area and circularity were calculated by tracing the droplet perimeter using the wand tool, then measuring using the Measure command.

## 2.5 Rheometry

Rheological analysis was performed using a Kinexus rheometer (Malvern Instruments, Worcestershire, UK) using a cone and plate geometry, with a gap height of 0.14 mm. Yield stress analysis was performed by conducting amplitude sweeps, on oscillatory mode with a frequency of 1 Hz. Thixotropic analysis of granular gels was conducted by carrying out a custom-built isothermal, stress-controlled, single frequency oscillation test. The test was carried out for 4.5 min, where the shear stress was alternated between low and high values–below and above the yield stress of the sample. The time at each stress was fixed at 30 s. Samples were equilibrated at either 20 or 37˚C for 5 min in the rheometer prior to testing. All measurements were performed in triplicate, except for the yield stress measurements at 37˚C where n = 2.

## 2.6 Mass flow rate determination

30% (w/v) poloxamer ink was loaded into a print cartridge and equilibrated at 30˚C for 10 min prior to printing. The combined mass of ink, cartridge and nozzle was then measured pre- and post-extrusion. Mass flow rate (ṁ) was then determined using the equation $\dot{m} = \dot{Q} * \rho$, where $\dot{Q}$ is the volumetric flow rate and $\rho$ is the density of the ink.

## 2.7 Cell culture

Transduction of tumour cell lines, namely CFPAC-1 (pancreatic tumour cell line) and U87-MG (glioma cell line), with green fluorescent protein (GFP) and red fluorescent protein (RFP) respectively, and subsequent cloning of these cells was performed to obtain fluorescent cultures. CFPAC-1 cells were cultured in proliferation medium, Iscove's modified Dulbecco's medium 12440–053 (IMDM) (Gibco), supplemented with 10% foetal bovine serum F7524 (FBS) (Sigma/Merck, UK) and 1% Glutamax 100X (Gibco, UK), while U87-MG cells were cultured in MEM non-essential amino acid solution M7145 (Sigma/Merck, UK) supplemented with 10% FBS, 1mM sodium pyruvate (Gibco, UK), 1% Glutamax 100X and 1% MEM non-essential amino acid solution 100X. Standard culturing conditions of 37˚C and 5% $CO_2$ were used, and cells were passaged when the colonies reached 80% confluency (typically, every 2–3 days).

## 2.8 Fabrication of 3D printed structures

For printing of filament structures on tissue culture plates and within gellan gum granular gels, bioink cartridges were loaded with poloxamer ink, attached to a conical nozzle with an internal diameter of 210 μm and loaded in a Cellink BioX 3D bioprinter (Cellink, Sweden). A custom G-code, controlling extrusion speed and pressure was written in order to print filament structures. Droplet-based bioprinting was carried out by replacing the pneumatic printhead on the BioX with an electromagnetic droplet (EMD) printhead equipped with a 300 μm, straight nozzle. Mixing of the cellular component and a gelMA ink was carried out manually

using a pipette within the print cartridge followed by 30 s of vortexing the cartridge to ensure homogenous distribution of cells. The print cartridge was then loaded into the EMD print-head. All droplet printing was carried out using a print temperature of 37˚C. Following printing, the suspension medium was diluted at a 1:1 ratio with proliferation medium that had been supplemented with 2% antibiotic-antimycotic solution. Printed constructs were then incubated at 37˚C in 5% $CO_2$ for a desired amount of time.

Printing of helices in photo-curable suspension media was carried out, by loading bioink cartridges with FITC-gelMA ink, attached to a 27 G tapered nozzle. The photo-curable suspension medium was incubated and held at 37˚C. The FITC-gelMA ink was incubated at 20˚C for 15 min in the Cellink BioX printer prior to printing. Printing of helices was carried out using the printing parameters; pressure = 160 kPa, print speed = 20 mm·min$^{-1}$. The helices were printed using a custom G-code written in-house. Following printing the entire construct was irradiated with a 405 nm light source (3D Printer UV Resin Light, FGRYB Amazon, UK) for 10 min.

### 2.9 Mechanical testing

Photo-curable suspension media were photo-crosslinked in custom-made, cylindrical Teflon moulds with diameter of 3 mm and thickness of 1.6 mm. GelMA precursor solutions were injected into the Teflon moulds at 37˚C. Photo-crosslinking time was fixed at 10 min using a 405 nm light source. Discs were inspected for any air bubbles and those, which contained bubbles were excluded, n = 5 discs were tested for each gelMA concentration. GelMA-only controls were prepared in the same manner, only using LAP-supplemented PBS without gellan gum particles to dissolve gelMA at known concentrations. Compression testing was performed on the hydrogel discs, on a DMA Q800 dynamic thermal mechanical analyser (TA instruments) using a compression clamp set up. The discs were swollen in PBS at room temperature overnight before testing. Strain ramps were performed on the samples, by applying 30% strain·min$^{-1}$ from 0–90% strain. The Young's modulus was taken as the slope of the stress-strain curve from 5 to 10% strain. Data was processed in Microsoft Excel and OriginLab OriginPro 2018b.

### 2.10 Statistical analysis

Analysis of numerical datasets was performed using Minitab 18 software (Minitab LLC, State College, PA, USA). Statistical comparisons between two experimental groups were performed using two-tailed Student's t tests. The threshold for statistical significance was set to p = 0.05. Exact p values are provided in Figures.

## 3. Results

### 3.1 Granule production and gel rheological characterisation

The simultaneous shearing and cooling of dilute gellan gum solutions in the presence of monovalent ions consistently resulted in suspensions of gel particles that behaved as yield stress fluids. These gels when left undisturbed were found to resist flow, whilst being easily displaced by pipetting or stirring. In contrast, non-sheared *control* gels were solid, could not be stirred or pipetted, and fractured irreversibly at higher loads. The gel granules were predominantly irregularly shaped with spindle-like protrusions extending from the granule body (Fig 1a–1c). The gels obtained from this shearing methodology exhibited high granule interconnectivity in their undiluted (jammed) state (Fig 1a). Upon dilution, granular gels broke up into smaller clusters (Fig 1b and 1c). Although it was unclear if and when single granules were

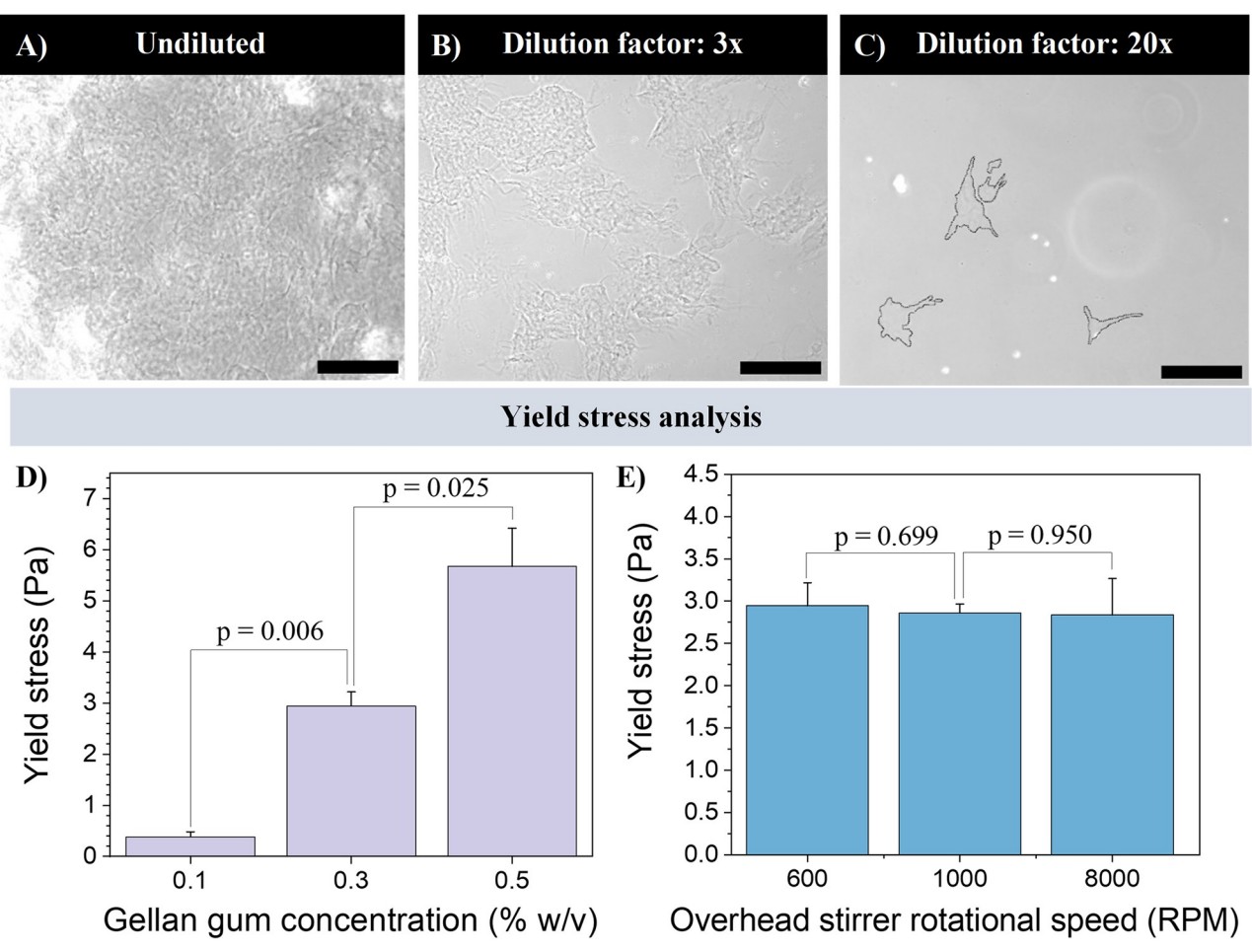

**Fig 1. Assessment of gellan gum granular gels.** (A-C) Brightfield micrographs of a 0.3% (w/v) gellan gum granular gel. (A) Undiluted gellan gum granular gel. (B-C) Gellan gum granular gels diluted with PBS using various dilution factors. (scale bars: 200 μm). (D-E) Yield stress dependence on the granular gel preparation conditions, where the yield stress was assessed as a function of either gellan gum polymer concentration (D) or the rotational speed of the overhead stirrer (E) n = 3.

obtained, it was evident that the granules had dimensions in the tens of microns. Stress amplitude sweeps on a rheometer, conducted at room temperature, revealed that the granular gels behaved as yield stress fluids, where the cross-over point of the storage modulus (G') and the loss modulus (G") was used to quantify the yield point. The yield stress depended strongly on gellan gum concentration (Fig 1d) but not on rotational speed of the overhead stirrer (Fig 1e) over the range of impeller speeds investigated.

Gellan gum granular gels continued to exhibit a yield stress when heated to 37°C (Fig 2), however a decrease from 3.4 ± 0.3 (n = 3) to 0.81 ± 0.07 Pa was observed (n = 2). Alternating between high and low stress confirmed reversibility of the solid-liquid transition (Fig 2b and 2d). The increase in incubation temperature of the granular gel was found to cause a slight decrease in the recovery time from 3.42 ± 0.15 s at 20°C to 3.25 ± 0.32 s at 37°C (n = 3), where the recovery time was determined as the time taken for G' to achieve 80% of its steady-state value [9]. G' recovery values were highly consistent after each successive high stress phase. The initial G' value during the first low stress phase differed markedly between each experiment, which is thought to relate to the operation of the rheometer and should not be taken as a fundamental difference in material property between the two temperatures.

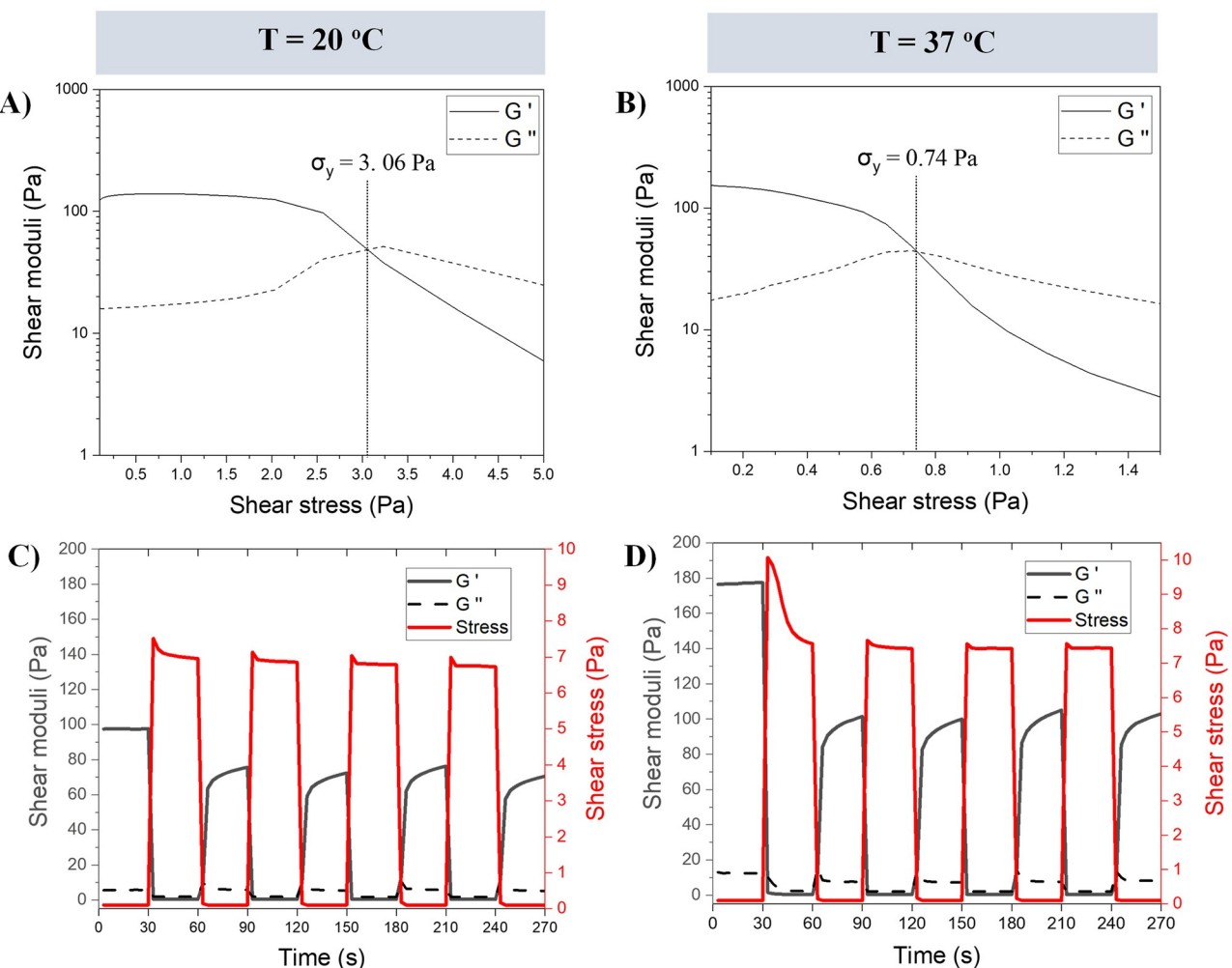

**Fig 2. Rheological analysis of a 0.3% (w/v) gellan gum granular gel performed at either 20˚C (A, C) or 37˚C (B, D).** (A-B) Representative steady shear stress ramp performed for yield stress (σy) determination. (C-D) Representative alternating low and high shear stress, (stress-controlled. single frequency oscillation test) performed to study self-healing behaviour of granular gels (n = 3).

## 3.2 3D printing in gellan gum suspension media

The suitability of the gellan gum granular gels for several bioprinting modalities was assessed, including extrusion printing of filaments and micro-valve bioprinting [17, 18]. Using poloxamer 407 as a model ink, continuous filaments were printed at various nozzle translational speeds using a conical nozzle with an inner diameter of 210 μm, either into gellan gum suspension media (GGSM), or onto a tissue culture polystyrene (TCPS) petri dish in air as a comparison (Fig 3). All prints were made using the same extrusion air pressure of 60 kPa, which resulted in the same mass flow rate of poloxamer ink extruded in either air onto TCPS, or into gellan gum granular gels (8.9 ± 0.4 mg·s$^{-1}$ vs. 8.6 ± 0.4 mg·s$^{-1}$, p = 0.551, n = 3). This indicates that the granular gel was not impeding flow of ink from the nozzle.

When printing into gellan gum suspension media, printing speeds as high as 60 mm·s$^{-1}$ still resulted in continuous printed filaments with minimal irregularities, whereas for printing onto TCPS, 10 mm·s$^{-1}$ was the maximum achievable speed before breaking up of filaments occurred (Fig 3).

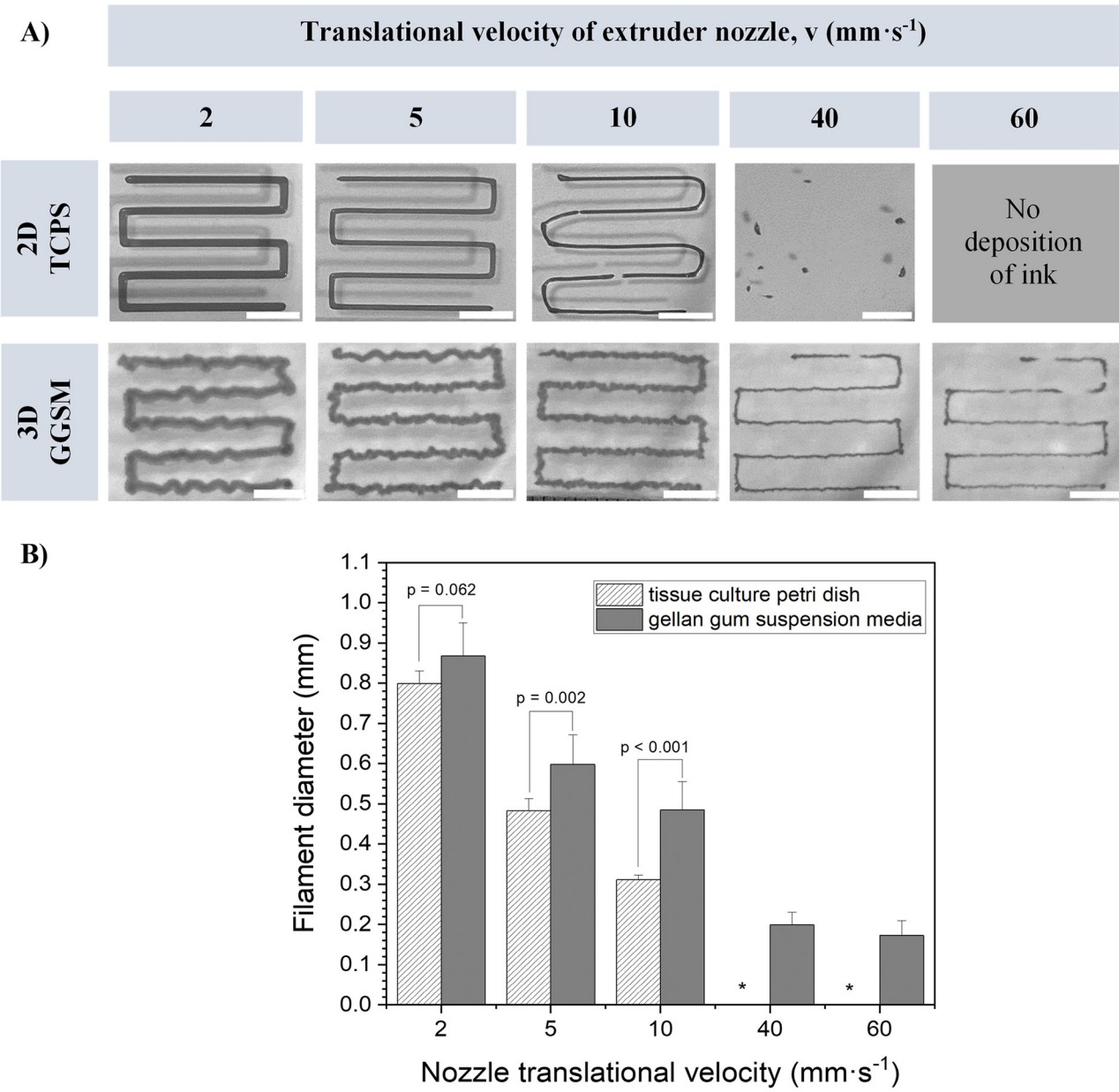

**Fig 3. Assessment of filaments printed with a poloxamer ink (30% w/v) onto/into different substrates.** (A) (top row) Filaments printed on tissue culture polystyrene petri dish. (bottom row) Filament printed into a 0.3% (w/v) gellan gum suspension medium. (scale bars: 5 mm). (B) Comparison of the effect of nozzle translational speed on the diameter of filaments printed in either gellan gum suspension media or on a tissue culture petri dish. Printing parameters: pressure = 60 kPa, print temperature = 30°C, nozzle = 210 µm conical nozzle (n = 10). Filaments printed in (A) the distance between the nozzle's tip and the substrate was calibrated to be the thickness of a piece of standard 80 gsm paper (ca. 0.1 mm). * no deposition of filaments.

During direct extrusion-based printing onto a solid substrate, the balance between the ink extrusion velocity and nozzle translational velocity as well as the distance between the nozzle orifice and substrate (stand-off distance), determines whether straight lines with a constant diameter can be printed [19]. We observe that as nozzle translational velocities (where stand-off distance is constant) are decreased, filaments printed onto TCPS maintained their

straightness but increased in diameter. In contrast, when printing into the suspension media, decreasing nozzle translational velocities resulted in the deposition of tortuous filaments (alongside increased diameters). This is due to the meandering instability of viscous threads [20] which in air is typically only seen for printing with fluids with viscosities that are orders of magnitude higher, such as in the electrospinning of polymer melts [21]. Similar to conventional printing onto TCPS, filaments printed in the suspension media exhibited a decrease in diameter with increasing translational speed at a constant flow rate (Fig 3b). For example, as nozzle translational velocity was increased from 2 mm·s$^{-1}$ to 40 mm·s$^{-1}$, filament diameter decreased from 0.87 ± 0.08 mm to 0.20 ± 0.03 mm, respectively (n = 10). Comparison of diameters of between filaments printed in suspension media and those printed on TCPS revealed a slight increase in diameter for filaments printed in the media for equivalent translational speeds. It should be noted here that all printing was carried out using a conical nozzle, which requires a lower pressure to obtain similar flow rates of ink in comparison to straight nozzles (S1 Fig).

Next, gelMA-based bioinks were jetted into gellan gum granular gels using an electromagnetic droplet printing jet, which combines air pressure with the open-close action of a magnetic ball valve. GelMA droplets were deposited using various extrusion pneumatic pressures in suspension media incubated for 30 min at various temperatures prior to printing, and droplet size and circularity were studied (Fig 4). Droplets were found to exhibit an elongated,

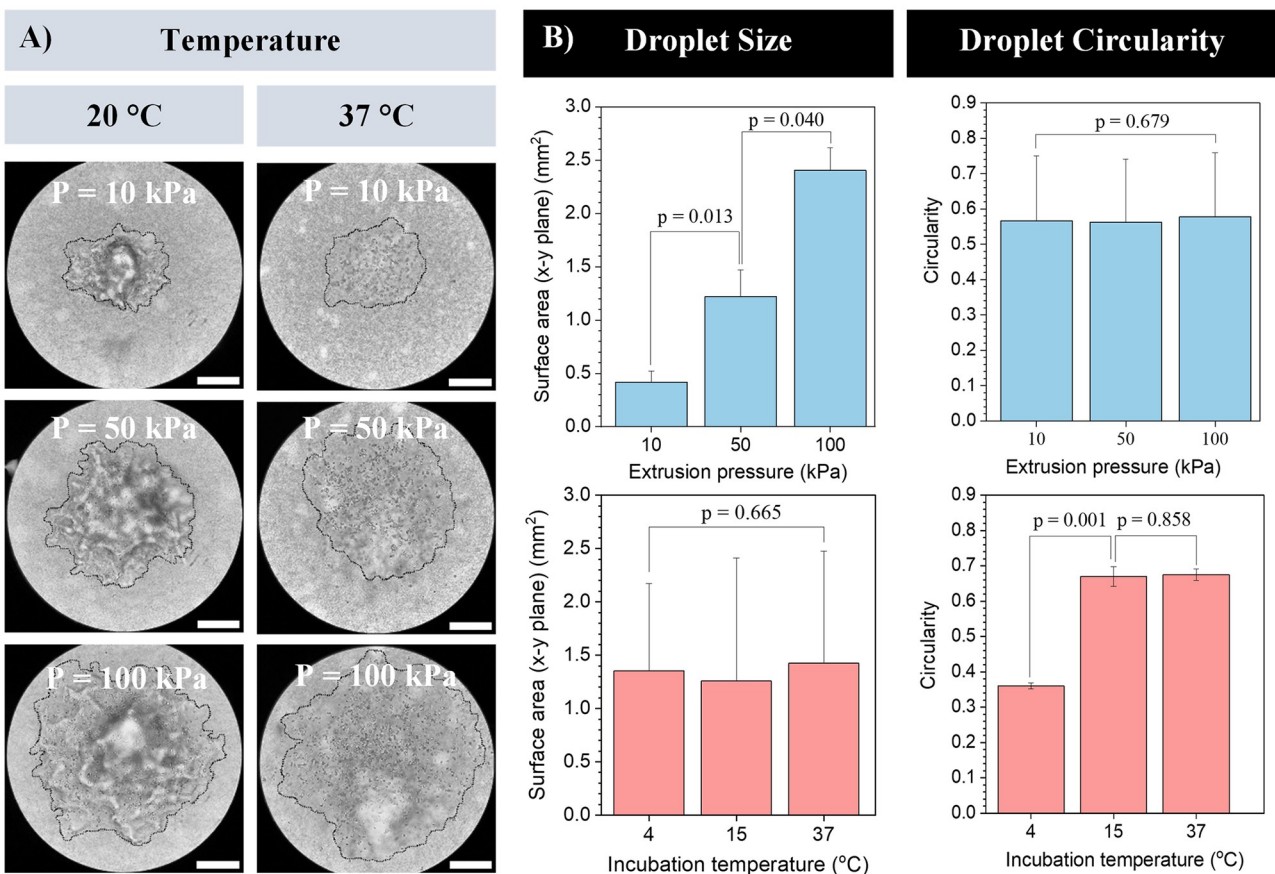

**Fig 4. Droplet bioprinting into gellan gum suspension media at various media temperatures.** Ink/cartridge temperature = 37˚C, nozzle = 300 μm straight nozzle. (A) Bright field micrographs of 10% (w/v) gelMA bioink droplets. (scale bars: 500 μm). (B) Effect of extrusion pressure and granular gel incubation temperature on either the droplet size or the droplet circularity (n = 3).

bullet-like shape (S2 Fig). Droplet size depended strongly on extrusion pressure, but not on the temperature of the gellan gum granular gel in the range studied (Fig 4b). Conversely, the circularity of the droplet was found to be independent of extrusion pressure (around 0.57 for all pressures, n = 9), but showed a degree of dependence on the temperature of the granular gel. Droplet circularity was defined by the equation $4\cdot\pi\cdot(\text{area}/\text{perimeter}^2)$ with a value of 1.0 indicating a perfect circle. Varying the temperature of the gellan gum suspension media revealed that at 4°C, values of circularity for the droplets were low, around 0.36. Circularity was higher at 15 and 37°C, with no significant difference in circularity between these temperatures.

Printed gelMA droplets laden with fluorescent CFPAC-1 GFP human pancreatic cancer cells are shown in Fig 5a. To achieve this, it was necessary to supplement the granular gel medium with 0.05 wt% LAP photo-initiator prior to printing, to prevent the loss of LAP from the ink droplet into the surrounding medium through diffusion. CFPAC-1 GFP cells maintained high fluorescence intensity from Day 0 to Day 1 in culture. Fluorescence intensity did decrease over 7 days of cell culture, which could have a variety of causes as elaborated on in the Discussion.

Additionally, constructs were fabricated containing two distinct zones each with a different cell population (Fig 5b). Two gelMA bioink formulations, containing either CFPAC-1 GFP or U87-MG Cherry cells, were deposited subsequently next to one another in granular gels, pre-incubated at either 10 or 37°C. These temperatures span across the melting temperature of the gelMA component of the ink, which appeared to have a critical impact on the ability of the droplets to fuse into a single construct. At a printing temperature of 10°C the droplets dispersed upon subsequent dilution of the suspension media, while at the higher printing temperature of 37°C the droplets were permanently fused. This indicated fusion of multi-material droplets was prevented by thermal gelation of the ink. This underscores the importance of the ability to use the suspension media at different temperatures.

## 3.3 Photo-curable suspension media

Next, we explored the possibility of rendering the gellan gum suspension media photo-curable through addition of gelMA. GelMA was added at 5–10 wt% to a 0.3 wt% gellan gum granular gel and used at 37°C. Using a fluorescently labelled gelMA (10% w/v) ink, a non self-supporting helix structure was printed successfully in the media with or without added gelMA (Fig 6). Post printing, the whole construct was exposed to 405 nm light. When using only gellan gum in the suspension medium, only the printed feature was crosslinked and thus could be liberated from its immobilised state upon dilution with PBS. However, when the suspension medium contained gelMA as well, both the ink and suspension media were crosslinked. The complete construct was then removed from the beaker which held the volume of the suspension media (Fig 6b) and could be incubated at 37°C without melting of either the construct or the printed features occurring.

Photo-curable suspension media could be created using different gelMA concentrations, resulting in different final mechanical properties (Fig 6c). Young's modulus, a key determinant of the functionality of cells cultured within 3D hydrogels for tissue engineering [22], ranged from 0.9–9.3 kPa for neat gelMA gels at concentrations between 5 and 10%, respectively. The presence of 0.3% w/v gellan gum granules roughly doubled the stiffness compared to neat gelMA, as the photo-curable suspension media exhibited Young's moduli values in the range of 1.9–17.5 kPa. For both conditions, the modulus (E) followed a power law (as is typically observed for such gels [23]) with a similar exponent value: $E = 0.0113\cdot c^{3.1739}$ for gels prepared

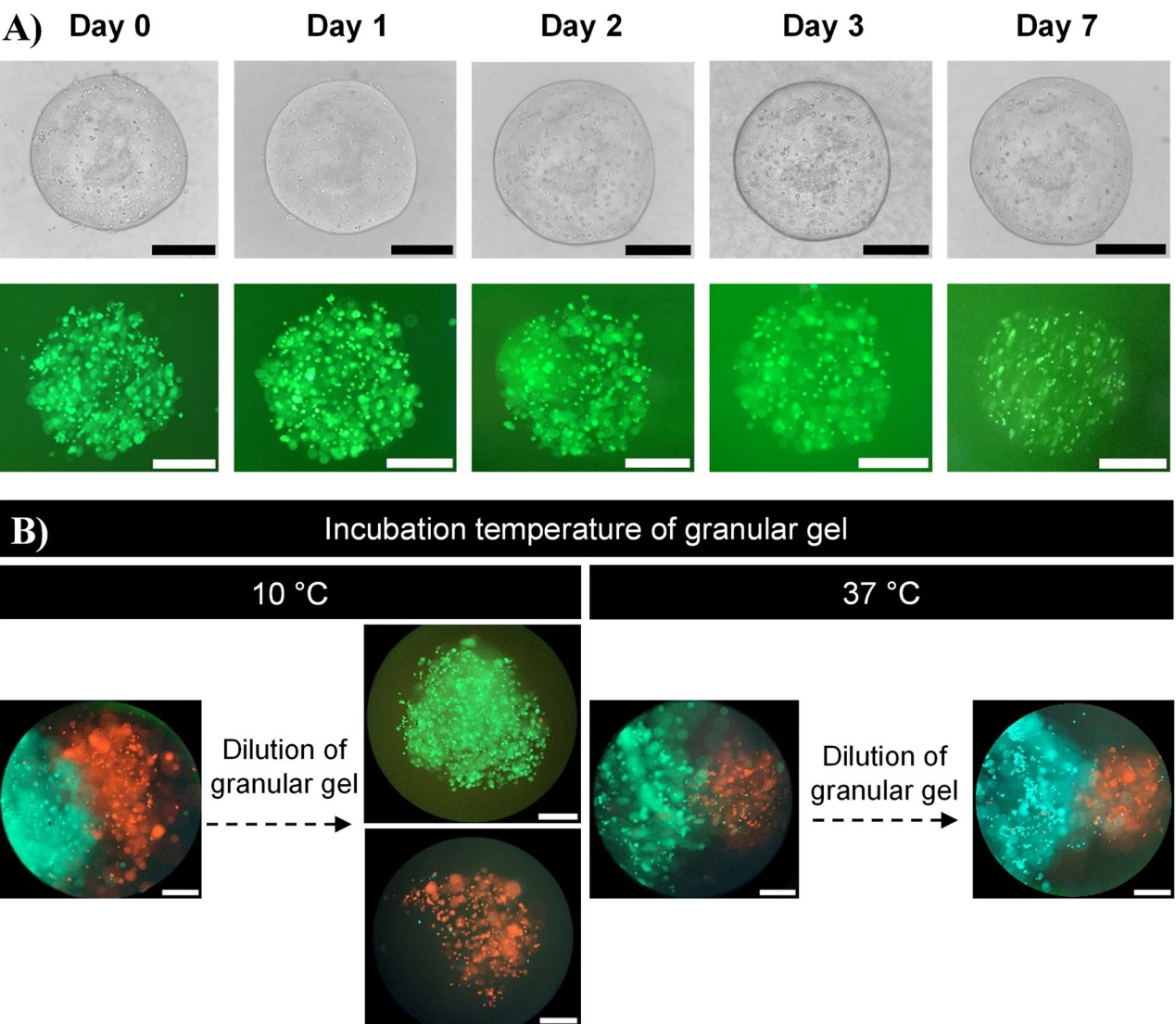

**Fig 5. Cell-laden bioink droplets printed into gellan gum suspension media.** (A) Brightfield and fluorescent images of a bioink droplet of 10% w/v gelMA + 0.05% w/v LAP laden with 3.5·106 CFPAC-1 GFP mammalian cells·mL-1 (scale bars: 500 μm). Droplet was incubated at 37˚C for 7 days and was shown to remain stable and not completely dissolve. (B) Suspension media incubation temperature (10 or 37˚C) was varied to show that cell-laden gelMA bioink droplets could be fused together using the temperature of the suspension medium. Bioinks were formulated from 10% w/v gelMA + 0.05% w/v LAP with either CFPAC-1 GFP (green) or U87-MG RFP (red) at 2·106 cells·mL-1 (scale bars: 500 μm).

with gelMA and gellan gum granular gel, and $E = 0.0035 \cdot c^{3.3934}$ for neat gelMA hydrogels, where c is the gelMA concentration (% w/v).

## 4. Discussion

In this work, we present a methodology for facile production of gellan gum granular gels for use as a suspension media for biofabrication applications. Through the application of a shearing force during the sol-gel transition of gellan gum solutions containing monovalent ions, we were able to rapidly create dense suspensions of gellan gum granules rather than a bulk, solid gel. We observe these granules to be highly irregular in shape, similar to the agarose granules

**A)** **GelMA-FITC hydrogel printed in a gellan gum suspension medium**

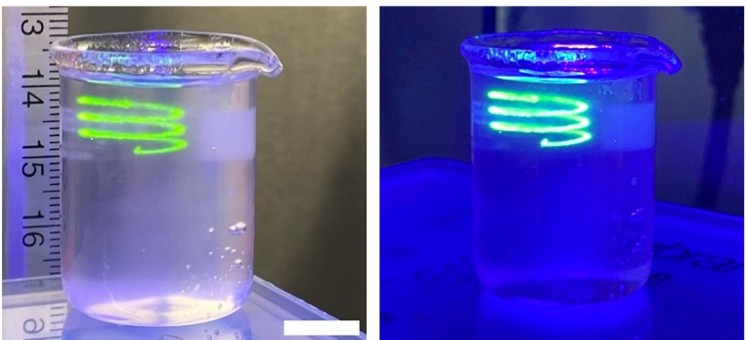

**B)** **GelMA-FITC hydrogel printed in a crosslinkable suspension medium**

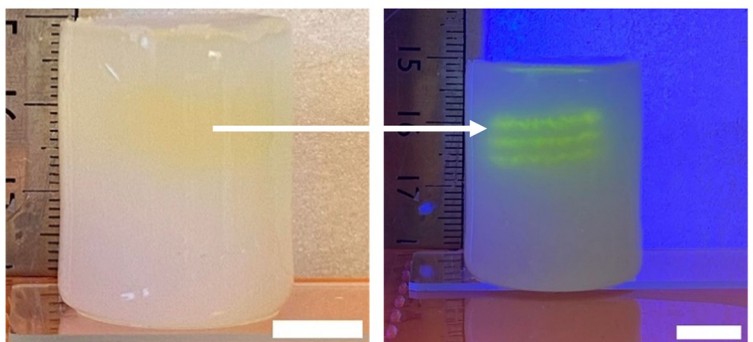

**C)**

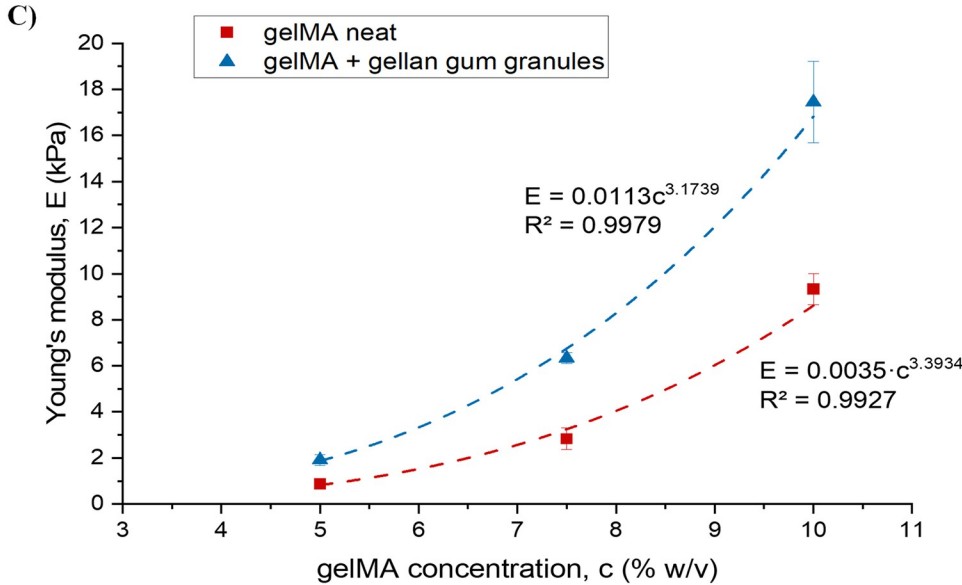

**Fig 6. Crosslinkable suspension medium consisting of gelMA and gellan gum granular gel.** (A) 3D printed, non-self-supporting helix printed using a fluorescein isothiocyanate (FITC) labelled gelMA hydrogel ink, printed in a gellan gum granular gel, in daylight (left) and 405 nm UV light (right). (scale bar: 10 mm). (B) Photo-cured hydrogel construct, where gelMA-FITC hydrogel helix printed in a gelMA and gellan gum suspension medium, in daylight (left) and 405 nm UV light (right). (scale bars: 10 mm). (C) Comparison of Young's modulus of chemically crosslinked gelMA hydrogels and gelMA + gellan gum granular gel (i.e. photo-curable suspension media) n = 5.

developed by Senior *et al*. [24], with the spindles extending from the granule body promoting a high degree of interlinking between granules. The granular gels behaved as yield stress fluids and exhibited time-dependent recovery of their solid-like properties after fluidisation and displacement of granules under the application of stress. We assessed the applicability of these gels for use with common biofabrication techniques such as extrusion 3D printing and microvalve bioprinting, finding that the morphology of printed inks was greatly affected by the surrounding granular gel.

When printing into suspension media, the ink's temperature will quickly equilibrate to the suspension media temperature due to the large difference in volume. The temperature-dependency of suspension media is therefore critical for successful suspended printing, and possibly limits the range of inks that can be used in combination with it. For example, a gelatine-based ink combined with a gelatine-based granular gel support bath (necessarily below gelatine's melting temperature) would quickly solidify and clog the nozzle. We demonstrated that gellan gum granular gels exhibit yield stress fluid behaviour over a range of temperatures, having yield stress values around 0.7 Pa at 37˚C, 3 Pa at 20˚C and exhibiting the capability to suspend droplets at temperatures as low as 4˚C. We propose that, as such, these granular gels offer the opportunity to accommodate printing of a wide variety of ink formulations, demonstrating appropriate rheological behaviour over the full bioprinting temperature window. Furthermore, the existence of a yield stress at 37˚C indicates that these granular gels can be used under cell culture conditions without loss of viscoplasticity. The yield stress values of our granular gels were observed to be lower than those for commonly used suspension media in the field, for example, Carbopol or alginate microparticle suspensions which exhibit yield stress values around 20–25 Pa [6] and 316 Pa [25] respectively, also measured at 1 Hz. The gellan gum granular gels presented here are therefore considered to fluidise with greater ease under the application of stress. Ease of fluidisation of the suspension media is advantageous for bioprinting applications as minimising resistance on the extruded ink as it flows from the nozzle.

A comparison between filament morphology and diameter when printing in our suspension medium versus printing on a TCP substrate, showed that filament morphology tended towards one of higher tortuosity while filament diameter increased when printing was conducted in the suspension medium. This comparison was performed over a range of nozzle translational speeds. While it has been previously discussed that filament deposition can be affected by ink rheology, for example viscoelasticity and yield stress [19, 26], these results highlight the importance that the print substrate, environment surrounding the nozzle and the relationship between the nozzle translational speed and the velocity of the extrudate leaving the nozzle, all contribute towards the resolution of a deposited filament. Filaments printed in air exhibited a small but significant reduction in diameter with increasing translational nozzle speed and maintained a smooth morphology, typical for hydrogels printed in air on a solid substrate [27]. In this case, filament morphology and diameter may be attributed towards several phenomena. It is likely that smoothing of the filament is partially the result of surface tension occurring at the air-ink interface with a rise in the Laplace pressure acting on the extrudate [28]. An additional contributing factor is likely the effects of extrudate swell (Barus effect), where there is release of the compressive forces that cause the contraction of hydrogel polymer chains as the extrudate leaves the nozzle orifice [9]. The morphology, including diameter, of the filament may also be influenced by the ink-substrate contact angle, where angles < 90˚ can cause large deformations and spreading of the hydrogel ink on the substrate [28]. Meanwhile, the observed decrease in filament diameter as nozzle translational speed is increased, likely relates to the printing speed exceeding the speed of material being extruded from the nozzle, causing the filament to become stretched [19].

While the resolution of printed filaments in air is the summation of a number of different phenomena, printing in a suspension medium significantly influences their contributions to affecting the morphology of the filaments. Printing in our gellan gum granular gels, which primarily constitute water, will significantly reduce the action of surface tension forces on the printed filaments thus inhibiting filament contraction due to Laplace pressure. This meandering instability of viscous threads is very well described in literature [20], including in the context of suspended extrusion printing [29] and is governed by the viscoelastic properties of both the suspension bath and ink. Similarly to printing in air, the decrease in filament diameter with increasing print speed, suggests the hydrogel ink may be being anchored to the gellan granules, causing stretching and thinning of the filament. The tortuosity we observe when printing in these granular gels may be exploited for engineering tissue structures where a high degree of folding is present in the native architecture for example in brain tissue or the epithelium of the small intestine [30, 31].

Following on, we assessed the applicability of gellan gum granular gels to micro-valve jetting—an important technique for controlling the spatial organisation of distinct small cell populations, e.g., for *in vitro* tissues for high-throughput drug testing. Droplets deposited in air are mostly spherical being driven primarily by surface tension, however droplets printed in our granular gels were found to adopt an elongated, cylindrical shape. In the absence of high interfacial forces that favour spherical droplet formation, the volume of ink ejected from the nozzle tip maintains a geometry closer resembling that prior to being ejected from the nozzle. We observed some swelling of droplets where their projected surface area was higher than the surface area of the nozzle orifice, which is a common occurrence when using pneumatic based printing systems [9]. This macroscopic swelling is driven by the phenomenon known as the Barus effect, mentioned previously. As such, it was anticipated droplets would exhibit high surface areas. More interestingly, the circularity of droplets showed a dependence not on extrusion pressure but on the temperature of the granular gel. We hypothesise that the irregular droplet surface is caused by the ink being pushed into the interstitial space between the gellan gum granules as the volume of ink is injected. At 4˚C, rapid heat loss from the small ink droplet (at 37˚C) to the cold surrounding media causes fast gelation and fixation of the irregular shape, resulting in low circularity values. At 37˚C, the interfacial forces act to cause a degree of relaxation of the ink into a spherical shape as the volume of gelMA ink doesn't solidify at that temperature. With no significant change between circularity values of droplets at 15 and 37˚C, it could be considered that at 15˚C the action of any interfacial forces on the ink is occurring fast relative to the gelation of the gelMA.

With current extrusion-based bioprinting strategies, the fabrication of clinically relevant volumes of tissue requires long print times, where incidences of cell-death are increased [32]. Here, we used a photo-curable suspension media to fabricate a construct with dimensions greater than several millimetres, utilising the addition of gelMA in our gellan gum granular gels. Photo-curing is commonly used for the encapsulation of cells in hydrogels [33]. Photocuring of the suspension media removes the need to extrude the bulk of tissue construct line-by-line, while the yield stress of the gellan gum granular gel permits the capability to engineer features/regions in the medium with compositions and properties that differ from the bulk medium. The Young's modulus was found to increase by about twofold when gelMA gels were prepared with gellan gum granular gels, which was roughly in line with earlier observations of photo-cured solutions of gelMA and gellan gum, though in that case the gellan gum was not in particle form [34]. This methodology for rapidly engineering large constructs presents a different option from methods such as casting of cell-laden gel as bulk tissues [35], with omnidirectional printing in the suspension media allows complex features to be deposited anywhere in the bulk gel.

## 5. Conclusion

In summary, gellan gum granular gels were developed using simultaneous shearing and cooling of polymer solutions through their sol-gel transition temperature in the presence of monovalent cations. These gels were developed to offer a method for depositing water-rich biomaterials in a 3D environment, thus alleviating some of the inherent limitations of direct extrusion 3D printing in air, where fabricating structures of high complexity that require mechanical support are not possible. Due to these gels exhibiting viscoplasticity at 37˚C, this enables bioprinting of cell-laden bioinks at optimum biological temperatures that are more permissive to cell survival. Furthermore, we discuss the implications of printing in these exceptionally low yield stress granular gels, detailing the challenges of predicting shape fidelity that arise from factors such as a reduction in surface tension compared to printing in air, and the ease of displacement of the granules by the nozzle and extrudate. Lastly, in response to the current challenge within the tissue engineering field of bioprinting large functional tissues, we looked at the feasibility of developing a photo-curable suspension media. To achieve this, the addition of gelMA to the gellan gum granular gels was utilised, showing the possibility to print gelMA inks within this photo-curable suspension media. We foresee this strategy could be exploited to embed cells or organoids and print surrounding vascular channels all within the same bulk suspension medium.

## Supporting information

**S1 Fig. Effect of nozzle selection on filament fidelity, when printing into a gellan gum fluid gel.** Both the conical and straight nozzles have an inner diameter of 25G. (scale bars: 5 mm). The straight nozzle permitted printing at a high pressure compared to the conical nozzle, negating the over-deposition of ink seen when printing with this nozzle. Printing parameters: ink formulation = 30% w/v poloxamer, print temperature = 30˚C. The distance between the nozzle's tip and the substrate was calibrated to be the thickness of a piece of standard 80 gsm paper.
(TIF)

**S2 Fig. Evaluation of typical droplet shape when extruded in a gellan gum granular gel.** Droplet printed with a gelMA-based bioink. The droplet was photocured in the granular gel, prior to unjamming of granular gel with culture media to permit rotation of the droplet and imaging in the z-plane. (scale bars: 500 μm).
(TIF)

**S1 Data.**
(ZIP)

**S2 Data.**
(ZIP)

**S3 Data.**
(ZIP)

**S4 Data.**
(ZIP)

## Author Contributions

**Conceptualization:** Andrew McCormack, Nicholas R. Leslie, Ferry P. W. Melchels.

**Data curation:** Andrew McCormack, Laura M. Porcza.

**Formal analysis:** Andrew McCormack, Laura M. Porcza, Ferry P. W. Melchels.

**Funding acquisition:** Nicholas R. Leslie, Ferry P. W. Melchels.

**Investigation:** Andrew McCormack, Laura M. Porcza.

**Methodology:** Andrew McCormack, Laura M. Porcza.

**Supervision:** Nicholas R. Leslie, Ferry P. W. Melchels.

**Visualization:** Laura M. Porcza.

**Writing – original draft:** Andrew McCormack, Ferry P. W. Melchels.

**Writing – review & editing:** Andrew McCormack, Nicholas R. Leslie, Ferry P. W. Melchels.

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
