## [Decision Letter · Decision Letter 0]

10 Sep 2024

PONE-D-24-32977Gellan gum-based granular gels as suspension media for biofabricationPLOS ONE

Dear Dr. Melchels,

Thank you for submitting your manuscript to PLOS ONE. After careful consideration, we feel that it has merit but does not fully meet PLOS ONE’s publication criteria as it currently stands. Therefore, we invite you to submit a revised version of the manuscript that addresses the points raised during the review process. Please submit your revised manuscript by Oct 23 2024 11:59PM. If you will need more time than this to complete your revisions, please reply to this message or contact the journal office at plosone@plos.org. Please include the following items when submitting your revised manuscript:A rebuttal letter that responds to each point raised by the academic editor and reviewer(s). You should upload this letter as a separate file labeled 'Response to Reviewers'.A marked-up copy of your manuscript that highlights changes made to the original version. You should upload this as a separate file labeled 'Revised Manuscript with Track Changes'.An unmarked version of your revised paper without tracked changes. You should upload this as a separate file labeled 'Manuscript'.

We look forward to receiving your revised manuscript.

Kind regards,

Pradeep Kumar, Ph.D.

Academic Editor

PLOS ONE

Journal Requirements:

1. When submitting your revision, we need you to address these additional requirements. Please ensure that your manuscript meets PLOS ONE's style requirements, including those for file naming. The PLOS ONE style templates can be found at https://journals.plos.org/plosone/s/file?id=wjVg/PLOSOne_formatting_sample_main_body.pdf and https://journals.plos.org/plosone/s/file?id=ba62/PLOSOne_formatting_sample_title_authors_affiliations.pdf.

 [AM EP/R513040/1 Engineering and Physical Sciences Research Council through their Doctoral Training Programme https://www.ukri.org/councils/epsrc/].  

[We acknowledge the financial support from the Engineering and Physical Sciences Research Council through their Doctoral Training Programme (EP/R513040/1)]

  [AM EP/R513040/1 Engineering and Physical Sciences Research Council through their Doctoral Training Programme https://www.ukri.org/councils/epsrc/]. 

Reviewers' comments:

Reviewer's Responses to Questions

**Comments to the Author**

1. Is the manuscript technically sound, and do the data support the conclusions?

Reviewer #1: Yes

Reviewer #2: Yes

2. Has the statistical analysis been performed appropriately and rigorously? 

Reviewer #1: Yes

Reviewer #2: Yes

3. Have the authors made all data underlying the findings in their manuscript fully available?

Reviewer #1: No

Reviewer #2: Yes

4. Is the manuscript presented in an intelligible fashion and written in standard English?

Reviewer #1: Yes

Reviewer #2: Yes

5. Review Comments to the Author

Reviewer #1: This paper presents a new method to create gellan gum based suspension media for extrusion bioprinting. The rheological properties of the obtained suspension baths were found favorable over a wide range of temperatures (4-37 degrees Celsius). This is a remarkable property because it enables to set the suspension bath's temperature to the optimal print nozzle temperature according to the requirements imposed by the bioink. Moreover, the study illustrates the use of the novel suspension media in the context of both extrusion-based printing and micro-valve bioprinting. In my opinion, this is a valuable contribution to the bioprinting literature and its impact could be further increased by addressing the following weak points:

1. The terms "droplet printing by jetting" (line 221), "electromagnetic droplet printing" (line 334), and "droplet jetting" (line 370) are confusing because they deviate from the professional jargon of bioprinting. Please use instead "micro-valve bioprinting", the common term described in detail in the following references (I am not a coauthor of them):

Gudapati H et al. A comprehensive review on droplet-based bioprinting: past, present and future. Biomaterials 2016, 102, doi:10.1016/j.biomaterials.2016.06.012.

Sun W et al. The bioprinting roadmap. Biofabrication 2020, 12, 022002, doi:10.1088/1758-5090/ab5158.

2. The description of extruded strand morphology versus nozzle translational velocity (printing speed) (lines 239-241) is accurate but too narrow. It is important to note that tortuous strands can result also in direct extrusion-based printing onto solid substrates provided that the stand-off distance is large enough and the printing speed is set well below the extrusion speed. The following work by Yuk and Zhao is an excellent study of strand morphology in direct ink writing:

Yuk H and Zhao X. A New 3D Printing Strategy by Harnessing Deformation, Instability, and Fracture of Viscoelastic Inks. Advanced Materials 2018, 30, 1704028, doi:10.1002/adma.201704028.

3. The list of printing parameters provided in the captions of Figures 3 and S1 should also include the stand-off distance (the distance between the nozzle's tip and the substrate).

4. In the Discussion, the authors suggest that the thinning of an extruded hydrogel strand with increasing printing speed occurs due to surface tension (lines 360-361). It is true, the Laplace pressure, of the order of hundreds of Pa in the case of the air-ink interface, contributes to smoothing the extruded strand. Nevertheless, the strand's diameter is determined by volume conservation, which explains why it is inversely proportional to the square root of the printing speed. The print resolution is also influenced by the extrudate swell effect (also known as the Barus effect or die-swelling): as the extruded hydrogel strand leaves the nozzle, its diameter becomes larger than the nozzle's orifice because of the relaxation of the hydrogel's polymer filament network. This effect is mentioned by the authors on line 379, in the context of droplet bioprinting. If the printing speed matches the dispensing velocity (extrusion speed) the filament is deposited as is (i.e. its diameter will be equal to the nozzle's inner diameter multiplied by the extrudate swell ratio). If the printing speed exceeds the extrusion speed, the filament is stretched on-the-fly (since it is anchored to the substrate by adhesion forces) and its diameter can be computed from volume conservation. Upon deposition onto a solid substrate (e.g. TCPS), it flattens to a certain extent, depending on the ink-substrate contact angle as well as on the ink's yield stress and shear thinning behavior.

I apologize for this long comment! I did not intend to lecture the authors; I just tried to illustrate how complex a problem is to predict the resolution of extrusion-based bioprinting. Theoretical studies are available in the literature for both direct and suspended extrusion bioprinting. The authors have the option to extend their discussion, providing a more accurate description of the observed fenomena, or to refrain from discussing the mechanisms responsible for their experimental findings.

5. Section 5 (Conclusion) is, in fact, a reiteration of the Abstract. In this part, it is fine to start with a brief summary of what has been achieved, but one should also attempt to explain the potential impact of the reported results and the perspectives of the proposed approach. Such comments are included the Discussion, but no take-home message is provided in the last section.

Minor remarks and revisions:

Line 19: I would stick to a single tense in the Abstract. It starts in simple present and switches midway to past tense. One option is to reformulate "Furthermore, printing of cell-laden droplets maintained over 7-day was demonstrated" as "Furthermore, we demonstrate the printing of cell-laden droplets maintained over 7 days".

Line 179: In the paragraph dedicated to statistical analysis, please state that "The level of statistical significance was set to p = 0.05."

Lines 223-224: Here, I would write "gellan gum suspension media (GGSM)" and "tissue culture polystyrene (TCPS)" (i.e. without 3D and 2D, respectively.) These are the acronyms used later on.

Line 252 and 257: Instead of "S1", please write "Supporting Information, Figure S1" and then, on line 257, please replace "S2" with "Figure S2".

Line 300: Instead of "Following printing of this structure," I would write "Post printing,".

Line 313: In the caption of Figure 6, I would keep "n = 5" as part of the sentence: instead of " suspension media). n =5" I would write " suspension media; n = 5)."

Line 316: Instead of "The Young’s moduli, a key determinant" consider writing "Young’s modulus, a key determinant".

Lines 321-323: Mathematical symbols (E, c) are usually typeset in Italic, using an equation editor.

Line 390: Instead of "extrusion-based printing", please write "extrusion-based bioprinting".

Line 397: Please replace "by approximately 2x" with "about twofold".

Line 408: Please consider revising "permitting the ability" as "allowing".

Reviewer #2: The manuscript discusses work pertaining to the suitability of granular gellan gum gels as suspension media for 3D bioprinting of cell-laden inks. The effects of gel concentration, temperature and rheology are described in terms of how such granular gels support 3D printed structures while achieving suitable printing speeds and resolutions. The manuscript is concise and well-written in a simple manner which makes it understandable and readable to various academic levels of audience. The rationale of the study is well-outlined in the introduction. The results support the hypothesis and reflect the objectives of the study. Below are some minor points to consider:

1. Line 19: amend to " 7 days"

2. Line 58: "suspension media"

3. Reference should be included for lines 67-68

4. Suggested to include a brief definition or explanation of "jammed granular network" in the introduction.

5. In materials and methods, it is suggested to include the concentrations of gellan gum and gelMA used, including the ratios of gellan gum:gelMA for the photo-curable medium.

5. Please include the time duration of shearing of the gellan gum gels at 600 rpm.

6. Line 127: replace full stop with a comma between "stress-controlled" and "single frequency.

7. Line 149: "G-code" include hyphen.

8. Line 169: amend to ",...which contained bubbles were excluded, n = 5..."

9. Line 367: amend to "brain tissue or of the epithelium of..."

6. PLOS authors have the option to publish the peer review history of their article (what does this mean?). If published, this will include your full peer review and any attached files.

Reviewer #1: No

Reviewer #2: No

---

## [Author Response · Author response to Decision Letter 0]

24 Sep 2024

We would like to thank the reviewers for their thoughtful comments. Please find below our point-by-point response to the reviewer reports for our manuscript. 

Reviewer #1

1. The terms "droplet printing by jetting" (line 221), "electromagnetic droplet printing" (line 334), and "droplet jetting" (line 370) are confusing because they deviate from the professional jargon of bioprinting. Please use instead "micro-valve bioprinting", the common term described in detail in the following references (I am not a coauthor of them)

Response: We have updated these terms as suggested by the reviewer, including the references to provide additional information to the readers on micro-valve bioprinting. 

2. The description of extruded strand morphology versus nozzle translational velocity (printing speed) (lines 239-241) is accurate but too narrow. It is important to note that tortuous strands can result also in direct extrusion-based printing onto solid substrates provided that the stand-off distance is large enough and the printing speed is set well below the extrusion speed. The following work by Yuk and Zhao is an excellent study of strand morphology in direct ink writing

Response: We would like to thank the reviewer for highlighting there are additional printing parameters which can affect printed strand morphology aswell as providing a reference - it is indeed some interesting work that we find worthy of a mention in our manuscript. We have rephrased some of the text to provide the reader with a more nuanced understanding of printing straight lines using direct extrusion-based printing, while including this reference. See below, 

(lines 250-254) During direct extrusion-based printing onto a solid substrate, the balance between the ink extrusion velocity and nozzle translational velocity aswell as the distance between the nozzle orifice and substrate (stand-off distance), determines whether straight lines with a constant diameter can be printed (19). We observe that as nozzle translational velocities (where stand-off distance is constant) are decreased, filaments printed onto TCPS maintained their straightness but increased in diameter. 

3. The list of printing parameters provided in the captions of Figures 3 and S1 should also include the stand-off distance (the distance between the nozzle's tip and the substrate).

Response: In these experiments, manual calibration was conducted where the distance between nozzle tip and substrate was the thickness of a piece of standard 80 gsm paper (around 0.1mm). To inform readers of this we have adapted the captions in both Figures to include this information stating, “the distance between the nozzle's tip and the substrate was calibrated to be the thickness of a piece of standard 80 gsm paper (ca. 0.1 mm).”

4. In the Discussion, the authors suggest that the thinning of an extruded hydrogel strand with increasing printing speed occurs due to surface tension (lines 360-361). It is true, the Laplace pressure, of the order of hundreds of Pa in the case of the air-ink interface, contributes to smoothing the extruded strand. Nevertheless, the strand's diameter is determined by volume conservation, which explains why it is inversely proportional to the square root of the printing speed. The print resolution is also influenced by the extrudate swell effect (also known as the Barus effect or die-swelling): as the extruded hydrogel strand leaves the nozzle, its diameter becomes larger than the nozzle's orifice because of the relaxation of the hydrogel's polymer filament network. This effect is mentioned by the authors on line 379, in the context of droplet bioprinting. If the printing speed matches the dispensing velocity (extrusion speed) the filament is deposited as is (i.e. its diameter will be equal to the nozzle's inner diameter multiplied by the extrudate swell ratio). If the printing speed exceeds the extrusion speed, the filament is stretched on-the-fly (since it is anchored to the substrate by adhesion forces) and its diameter can be computed from volume conservation. Upon deposition onto a solid substrate (e.g. TCPS), it flattens to a certain extent, depending on the ink-substrate contact angle as well as on the ink's yield stress and shear thinning behavior.

Response: We would like to thank the reviewer for their very insightful and well explained comments on the variables which contribute towards diameter/smoothness of printed filaments. We find these comments to be useful and should be addressed in our manuscript. We have extended the discussion to include the following text. 

(line 369 – 400) A comparison between filament morphology and diameter when printing in our suspension medium versus printing on a TCP substrate, showed that filament morphology tended towards one of higher tortuosity while filament diameter increased when printing was conducted in the suspension medium. This comparison was performed over a range of nozzle translational speeds. While it has been previously discussed that filament deposition can be affected by ink rheology, for example viscoelasticity and yield stress (Paxton et al., 2016; Yuk & Zhao, 2018), these results highlight the importance that the print substrate, environment surrounding the nozzle and the relationship between the nozzle translational speed and the velocity of the extrudate leaving the nozzle, all contribute towards the resolution of a deposited filament. Filaments printed in air exhibited a small but significant reduction in diameter with increasing translational nozzle speed and maintained a smooth morphology, typical for hydrogels printed in air on a solid substrate (Webb & Doyle, 2017). In this case, filament morphology and diameter may be attributed towards several phenomena. It is likely that smoothing of the filament is partially the result of surface tension occurring at the air-ink interface with a rise in the Laplace pressure acting on the extrudate (Ning et al., 2020). An additional contributing factor is likely the effects of extrudate swell (Barus effect), where there is release of the compressive forces that cause the contraction of hydrogel polymer chains as the extrudate leaves the nozzle orifice (Cooke & Rosenzweig, 2021). The morphology, including diameter, of the filament may also be influenced by the ink-substrate contact angle, where angles < 90° can cause large deformations and spreading of the hydrogel ink on the substrate (Ning et al., 2020). Meanwhile, the observed decrease in filament diameter as nozzle translational speed is increased, likely relates to the printing speed exceeding the speed of material being extruded from the nozzle, causing the filament to become stretched (Yuk & Zhao, 2018). 

While the resolution of printed filaments in air is the summation of a number of different phenomena, printing in a suspension medium significantly influences their contributions to affecting the morphology of the filaments. Printing in our gellan gum granular gels, which primarily constitute water, will significantly reduce the action of surface tension forces on the printed filaments thus inhibiting filament contraction due to Laplace pressure. This meandering instability of viscous threads is very well described in literature (Morris et al., 2008), including in the context of suspended extrusion printing (Prendergast & Burdick, 2021) and is governed by the viscoelastic properties of both the suspension bath and ink. Similarly to printing in air, the decrease in filament diameter with increasing print speed, suggests the hydrogel ink may be being anchored to the gellan granules, causing stretching and thinning of the filament. The tortuosity we observe when printing in these granular gels may be exploited for engineering tissue structures where a high degree of folding is present in the native architecture for example in brain tissue or the epithelium of the small intestine (Brassard et al., 2021; Gjorevski et al., 2016). 

5. Section 5 (Conclusion) is, in fact, a reiteration of the Abstract. In this part, it is fine to start with a brief summary of what has been achieved, but one should also attempt to explain the potential impact of the reported results and the perspectives of the proposed approach. Such comments are included the Discussion, but no take-home message is provided in the last section.

Response: We would like to thank the reviewer for the attentiveness here and agreed that the conclusion requires improvement. We thus have taken steps to reword the conclusion. Please see below. 

(line 434-449) In summary, gellan gum granular gels were developed using simultaneous shearing and cooling polymer solutions through their sol-gel transition temperature in the presence of monovalent cations. These gels were developed to offer a method for depositing water-rich biomaterials in a 3D environment, thus alleviating some of the inherent limitations of direct extrusion 3D printing in air, where fabricating structures of high complexity that require mechanical support are not possible. Due to these gels exhibiting viscoplasticity at 37 °C, this enables bioprinting of cell-laden bioinks at optimum biological temperatures that are more permissive to cell survival. Furthermore, we discuss the implications of printing in these exceptionally low yield stress granular gels, detailing the challenges of predicting shape fidelity that arise from factors such as a reduction in surface tension compared to printing in air, and the ease of displacement of the granules by the nozzle and extrudate. Lastly, in response to the current challenge within the tissue engineering field of bioprinting large functional tissues, we looked at the feasibility of developing a photo-curable suspension media. To achieve this, the addition of gelMA to the gellan gum granular gels was utilised, showing the possibility to print gelMA inks within this photo-curable suspension media. We foresee this strategy could be exploited to embed cells or organoids and print surrounding vascular channels all within the same bulk suspension medium. 

Minor remarks and revisions:

Response: These minor revisions were very helpful to improving the quality of the manuscript and have been addressed in the manuscript. 

Reviewer #2

We would like to thank the reviewer for their comments and detail-orientated focus on the manuscript – we believe this to have helped improve the quality of the work. All of the suggested revisions have been revised, notably including a definition of jammed granular network.

---

## [Decision Letter · Decision Letter 1]

14 Oct 2024

Gellan gum-based granular gels as suspension media for biofabrication

PONE-D-24-32977R1

Dear Dr. Melchels,

We’re pleased to inform you that your manuscript has been judged scientifically suitable for publication and will be formally accepted for publication once it meets all outstanding technical requirements.

Kind regards,

Pradeep Kumar, Ph.D.

Academic Editor

PLOS ONE

Additional Editor Comments (optional):

Reviewers' comments:

Reviewer's Responses to Questions

**Comments to the Author**

1. If the authors have adequately addressed your comments raised in a previous round of review and you feel that this manuscript is now acceptable for publication, you may indicate that here to bypass the “Comments to the Author” section, enter your conflict of interest statement in the “Confidential to Editor” section, and submit your "Accept" recommendation.

Reviewer #1: All comments have been addressed

Reviewer #2: All comments have been addressed

2. Is the manuscript technically sound, and do the data support the conclusions?

Reviewer #1: Yes

Reviewer #2: Yes

3. Has the statistical analysis been performed appropriately and rigorously? 

Reviewer #1: Yes

Reviewer #2: Yes

4. Have the authors made all data underlying the findings in their manuscript fully available?

Reviewer #1: Yes

Reviewer #2: Yes

5. Is the manuscript presented in an intelligible fashion and written in standard English?

Reviewer #1: Yes

Reviewer #2: Yes

6. Review Comments to the Author

Reviewer #1: The authors have addressed all the concerns expressed in my previous referee report. I think this is a nice paper that might elicit the interest of the bioprinting community.

Reviewer #2: All comments have been addressed.

I believe this manuscript can be accepted for publication.

7. PLOS authors have the option to publish the peer review history of their article (what does this mean?). If published, this will include your full peer review and any attached files.

Reviewer #1: No

Reviewer #2: No

---

## [Editor Report · Acceptance letter]

17 Oct 2024

PONE-D-24-32977R1 

PLOS ONE

Dear Dr. Melchels, 

I'm pleased to inform you that your manuscript has been deemed suitable for publication in PLOS ONE. Congratulations! Your manuscript is now being handed over to our production team.

Kind regards, 

on behalf of

Prof. Pradeep Kumar 

Academic Editor

PLOS ONE